# Channel Independence Improves Out-of-Distribution Generalisation in Multivariate Time Series Classification

## Abstract

Robustness to distribution shift is a necessary property of machine learning models for their safe and effective deployment. However, deep learning models are susceptible to learning spurious features of the in-distribution (ID) training data that fail to generalise to out-of-distribution (OOD) data. Domain generalisation algorithms aim to tackle this problem, but recent studies have demonstrated that their improvement over standard empirical risk minimisation is marginal. We address this problem for multivariate time series classification (TSC), where it is standard practise to use feature extractor architectures that learn with channel dependence (CD), enabling cross-channel patterns to be learned. Inspired by recent success in time series forecasting, we investigate how channel independence (CI) impacts OOD generalisation in TSC. Our experiments on six time series datasets reveal that ID and OOD features exhibit significantly greater distributional divergence when learned with CD compared to CI. As a consequence, models that learn with CI are more robust to distribution shift, evidenced by smaller generalisation gaps (the difference between ID and OOD performance) across datasets. On datasets that have a stronger shift, OOD accuracy is substantially higher for CI than CD.

## 1 Introduction

In real-world applications, machine learning models are often required to make predictions on out-of-distribution (OOD) data that is different from the in-distribution (ID) data they were trained on. Models should be *robust* to distribution shift, maintaining performance on OOD data that is comparable to that on ID data (Taori et al., 2020). Unfortunately, this is not typical and large *generalisation gaps*, the difference between OOD and ID performance, are common (Gulrajani & Lopez-Paz, 2021; Gagnon-Audet et al., 2023). This has been attributed to models learning features of the training data that are *spurious*; they correlate with labels but do not capture a causal relationship (Geirhos et al., 2020; Gulrajani & Lopez-Paz, 2021). A model that relies on spurious features will fail to generalise effectively to OOD data where those features are absent.

In this work, we focus on improving OOD generalisation in time series classification (TSC). TSC plays an important role in a range of applications, including activity recognition (Zhang et al., 2022a), disease diagnosis and monitoring (Oh et al., 2020), and predictive maintenance (Carvalho et al., 2019). Distribution shift in time series data can arise from inherent intra- and inter-person variabilities, as well as differences in recording equipment (Gagnon-Audet et al., 2023). For the safe and effective deployment of machine learning models in real-world TSC applications, robustness under distribution shift is vital.

In tackling this problem, we consider the *domain generalisation* setting: models are trained on a set of *source* domains, which form the ID data, and OOD generalisation is then assessed on a set of unseen *target* domains. Gagnon-Audet et al. (2023) showed that across time series benchmarks, models trained with empirical risk minimisation (ERM) (Vapnik, 1991) exhibit substantial generalisation gaps, highlighting the need for effective domain generalisation algorithms in this area. However, they also showed that existing algorithms only marginally improve upon the OOD accuracy of ERM. Thus, there remains a clear need for approaches that effectively improve OOD generalisation in TSC.

Table 1: The generalisation gap in terms of accuracy for CD and CI for each time series dataset, as a measure of robustness to distribution shift (smaller is better).

|     | DSADS | HAR | MHEALTH | PAMAP | WESAD | WISDM |
|-----|-------|-----|---------|-------|-------|-------|
| CD  | 19.9  | 3.6 | 14.1    | 34.0  | 25.4  | 12.6  |
| CI  | 5.6   | 2.8 | 2.5     | 4.8   | 6.8   | 7.5   |

In many TSC applications, the time series are *multivariate*, composed of multiple channels of univariate time series. In OOD generalisation research with multivariate time series, it is standard practise to use a feature extractor architecture that learns with *channel dependence* (CD), where features are a function of multiple channels (Lu et al., 2023; Gagnon-Audet et al., 2023; Ozyurt et al., 2023; He et al., 2023). The rationale is that richer features of the time series may be learned by considering cross-channel relationships, potentially improving task performance. The alternative is *channel independence* (CI), where learning is restricted to the individual channel, which has had recent success in time series forecasting (Nie et al., 2023; Zeng et al., 2023; Liu et al., 2024; Han et al., 2024). In this work, we propose a simple method for implementing CI for multivariate TSC, and investigate how these two types of learning impact OOD generalisation.

We find that learning with CD is often detrimental to OOD generalisation, and that learning with CI is a significantly more robust alternative, as shown in Table 1. Using six real-world multivariate time series datasets, we analyse ID classification performance and the distributional divergence between source (ID) and target (OOD) domain features, which bound performance in the target domain (Ben-David et al., 2006; 2010). Our results show that models that learn with CD outperform CI on ID data, but that the features learned exhibit far greater distributional divergence, indicating a tendency to learn spurious features. On datasets with more severe distribution shifts, the greater robustness of CI enables it to vastly outperform CD on OOD data.

We extend our work by exploring the use of the *frequency domain*[1], which describes the content of a signal at individual frequencies and has emerged as a means of improving OOD generalisation in TSC (He et al., 2023; Mohapatra et al., 2024), in the context of CD and CI. Our experiments show that frequency domain features exhibit consistently lower distributional divergence than time domain features, but at the cost of worse ID classification performance. Importantly, this trade-off has different implications for OOD generalisation with CD and CI. For CD, where divergence is high, frequency features improve OOD performance. However, since CI models already show low divergence with time domain features, frequency features offer no additional benefit. This underscores that strategies improving OOD generalisation in CD models may not be effective for CI models.

## 2 RELATED WORK

**Domain generalisation in time series classification.** Domain generalisation algorithms learn from a set of labelled source domains and are evaluated for OOD generalisation on unseen target domains. A number of general approaches have been proposed, such as domain invariant learning (Ganin et al., 2016; Arjovsky et al., 2020), meta-learning (Li et al., 2018) and data augmentation (Volpi et al., 2018; Li et al., 2021). Gulrajani & Lopez-Paz (2021) compared domain generalisation algorithms on image classification benchmarks and found that none outperformed ERM. More recently, Gagnon-Audet et al. (2023) performed a similar study for time series tasks with both general and time series-specific domain generalisation algorithms, and similarly found only marginal improvement over ERM. An explanation for this is that although existing algorithms can improve robustness, this is outweighed by an accompanying decrease in ID classification performance (Sener & Koltun, 2022). Distinct from existing methods, our work explores how different ways of learning from multiple channels impacts OOD generalisation.

**Channel independence in time series forecasting.** Recently, CI has shown superior performance over CD in time series forecasting with transformers (Nie et al., 2023; Zeng et al., 2023; Liu et al., 2024). To explain this, Han et al. (2024) show that models that learn with CI perform worse on

---
[1]We also use the word 'domain' in this context to be consistent with signal processing literature.

ID data due to reduced learning capacity, but are more robust to distribution shift than models that learn with CD, and that this trade-off favours CI for forecasting. Although relevant to our work, we cannot assume that these results transfer to TSC as the nature of the tasks are different, as are the characteristics of the data. Forecasting datasets typically have sampling frequencies measured in minutes, hours, days or weeks (Han et al., 2024), while TSC applications use sampling frequencies on the order of tens, hundreds, or thousands of samples per second, as evidenced by the datasets used in this paper. Furthermore, their analyses are specific to forecasting, while ours are based on domain adaptation theory, which better aligns with OOD generalisation research.

**Channel independence in time series classification.** CI has been applied to standard (i.e. non-OOD) TSC to an extent. Ruiz et al. (2021) use CI simply as a means of adapting models designed for univariate time series to multivariate data by building an ensemble of univariate classifiers, each trained on a different channel, though this was in the context of non-deep learning models. In deep learning for TSC, Zheng et al. (2014) used a separate convolutional neural network (CNN) for each channel of a time series (though the features from each were fused before classification, meaning the models are not truly independent). However, subsequent state-of-the-art model architectures opted to learn with CD instead (Fawaz et al., 2019; Foumani et al., 2024). Research on OOD generalisation in TSC have all made use of feature extractor architectures that learn with CD (Wilson et al., 2020; Ragab et al., 2023; Lu et al., 2023; Gagnon-Audet et al., 2023; Ozyurt et al., 2023; He et al., 2023; Mohapatra et al., 2024). It therefore remains to be seen whether the benefits observed in forecasting from using CI also apply to OOD generalisation in classification.

**Frequency features in time series tasks.** Converting a signal from the time domain into the frequency domain has long been popular for improving performance in TSC applications (Lima et al., 2019; Saeidi et al., 2021), and more recently in self-supervised representation learning (Yang & Hong, 2022; Zhang et al., 2022b). Most relevantly, He et al. (2023) used the approach for domain adaptation, a similar setting to domain generalisation but with unlabelled target domain data available during training. They separately encode the time and frequency domain of time series and concatenate the features for classification, which showed improved OOD performance compared to time features alone. These results were observed when using a feature extractor that learns with CD, hence we investigate whether the same benefits similarly extend to models that learn with CI.

# 3 PRELIMINARIES

## 3.1 PROBLEM SET-UP

We consider the domain generalisation setting for TSC. Let $\mathcal{X} \subset \mathbb{R}^{L \times N}$ denote the input space for multivariate time series with $N$ channels and sequence length $L$, $\mathcal{Z} \subset \mathbb{R}^m$ be the $m$-dimensional feature space, and $\mathcal{Y} = \{1, \ldots, C\}$ be the output space for a $C$-class classification task. Input data samples are $X \in \mathcal{X}$, features are $Z \in \mathcal{Z}$ and labels are $y \in \mathcal{Y}$.

We have a set of source domains $\mathcal{E}_{\text{train}}$ and a set of target domains $\mathcal{E}_{\text{test}}$, each with a labelled dataset $D_d = \{(X_i, y_i)\}_{i=1}^{n_d}$ sampled i.i.d. from a joint probability distribution $P_d(X, y)$. Each distribution is different, such that $P_d(X, y) \neq P_{d'}(X, y)$ for all $d \neq d'$ with $d, d' \in \mathcal{E}_{\text{train}} \cup \mathcal{E}_{\text{test}}$. All domains share the same input, feature, and label spaces.

The datasets are combined to form the overall training dataset $D_{\text{train}} = \cup_{d \in \mathcal{E}_{\text{train}}} D_d$ and test dataset $D_{\text{test}} = \cup_{d \in \mathcal{E}_{\text{test}}} D_d$. The objective is to train a model $f = h \circ g (= h(g(\cdot)))$ using $D_{\text{train}}$ that generalises to unseen domains in the set of all possible domains $\mathcal{E}_{\text{all}}$, where $g : \mathcal{X} \to \mathcal{Z}$ is a feature extractor, $h : \mathcal{Z} \to \mathcal{Y}$ is a classifier, and $\mathcal{E}_{\text{train}}, \mathcal{E}_{\text{test}} \subseteq \mathcal{E}_{\text{all}}$. We evaluate the performance of $f$ on $D_{\text{test}}$ to estimate generalisation to $\mathcal{E}_{\text{all}}$.

The shift between two joint probability distributions $d$ and $d'$ may be decomposed as $P_d(y|X)P_d(X) \neq P_{d'}(y|X)P_{d'}(X)$. A common simplifying assumption in OOD generalisation research is that distribution shift is the result of a covariate shift (Zhao et al., 2019; Liu et al., 2023), which states that $P_d(X) \neq P_{d'}(X)$ while $P_d(y|X) = P_{d'}(y|X)$. This is the assumption that we adopt moving forward.

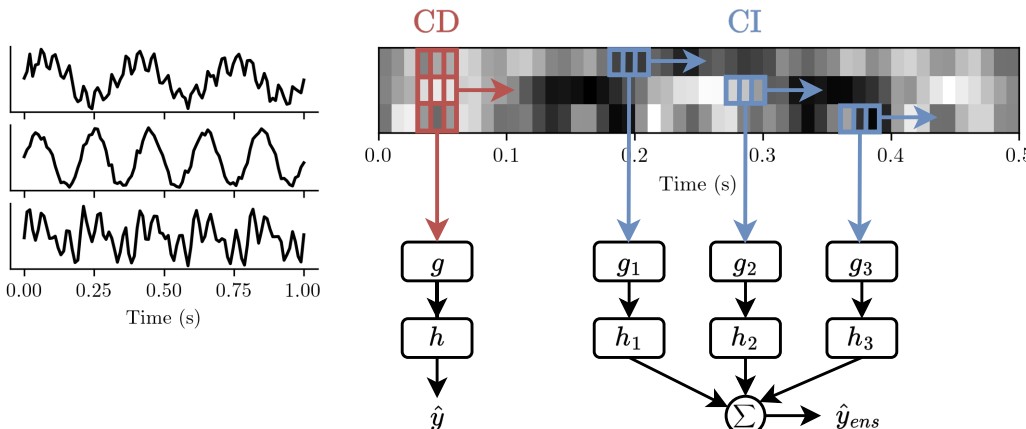

Figure 1: Learning from a three-channel time series with CD and CI using a 1D CNN as the feature extractor, represented with a single convolutional kernel (the grid structures). With CD, the model operates over all channels. In the channel-wise ensemble (for CI), each CNN only operates on a single channel.

## 3.2 Comparing Channel Dependence and Channel Independence

In this section, we clarify the difference between CD and CI with an example of a 1D CNN feature extractor (Ozyurt et al., 2023; He et al., 2023), illustrated in Figure 1. The same principles outlined here extend to other commonly used architectures, albeit with different formulations.

In a standard 1D CNN that learns with CD, the convolutional kernels in the first layer span all the channels of the input time series. Specifically, each kernel is parameterised by a matrix of height $N$, corresponding to the number of input channels, and a length $L_K$, determining the receptive field of the kernel. The kernels slide across the time dimension of a time series sample $X$, and perform a convolution at each step. Denote a single kernel $K \in \mathbb{R}^{N \times L_K}$, the output $o_i$ of the convolution between $K$ and $X$ at time step $i$ is:

$$o_i = \sum_{n=0}^{N-1} \sum_{l=0}^{L_K-1} X_{n,i+l} K_{n,l},$$

where $X_{n,i+l}$ is the value from the $n$th channel at time step $i + l$, and $K_{n,l}$ is the value from the $n$th row and $l$th column of the kernel. This formulation shows that the output at each time step is a function of all the channels in the time series. Since the parameters of $K$ are learnable, the model is able to capture cross-channel dependencies.

In contrast, learning with CI means that each channel of the time series is processed independently using convolutional kernels that are specific to that channel. Let $X^j \in \mathbb{R}^{1 \times N}$ be the $j$th channel of $X$, and $K^j \in \mathbb{R}^{1 \times L_K}$ be a kernel for channel $j$. The output $o_i^j$ for channel $j$ at time step $i$ is:

$$o_i^j = \sum_{l=0}^{L_K-1} X_{i+l}^j K_l^j.$$

The output at each time step depends only on channel $j$, making the learned features channel-specific.

Clearly, the kernel space of the CD approach $\mathcal{K} \subset \mathbb{R}^{N \times L_K}$ has greater capacity than the kernel space of the CI approach $\mathcal{K}^j \subset \mathbb{R}^{1 \times L_K}$. While this enables more complex patterns of the data to be learned with CD, the smaller capacity of CI provides implicit regularisation that may improve robustness.

## 4 LEARNING WITH CHANNEL INDEPENDENCE

### 4.1 CHANNEL-WISE ENSEMBLE

Models designed for univariate time series classification can be extended to multivariate time series by constructing an ensemble of models, each one trained on a different channel (Ruiz et al., 2021). This is the approach that we take for implementing CI, which we refer to as a *channel-wise ensemble*, illustrated in Figure 1.

We construct an ensemble of $N$ independently-trained models $\{f_j = h_j \circ g_j\}_{j=1}^{N}$ for multivariate time series with $N$ channels, where each model consists of a feature extractor $g_j$ and classifier $h_j$. In this work, we build a homogeneous ensemble, with the models in the ensemble having the same architecture. A heterogeneous ensemble could be used to boost performance if it is known that certain architectures are better suited to certain channels.

The $j$th model is trained on the dataset $D_{\text{train}}^{j} = \{(X_i^j, y_i)|(X_i, y_i) \in D_{\text{train}}\}$, which contains univariate time series from the $j$th channel of $D_{\text{train}}$. We use ERM (Vapnik, 1991) for training models in our experiments, which minimises the average loss over the training data:

$$f_j^* = \arg\min_{f_j \in \mathcal{F}} \hat{R}(f_j), \quad \hat{R}(f_j) = \frac{1}{|D_{\text{train}}^{j}|} \sum_{i=1}^{|D_{\text{train}}^{j}|} \ell(f_j(X_i^j), y_i),$$

where $\mathcal{F}$ is the hypothesis class for $f_j$, $\hat{R}(f_j)$ is the empirical risk, $|D_{\text{train}}^{j}|$ is the cardinality of $D_{\text{train}}^{j}$, and $\ell$ the cross-entropy loss. We use ERM given its simplicity and competitiveness with domain generalisation algorithms (Gulrajani & Lopez-Paz, 2021; Gagnon-Audet et al., 2023), though the channel-wise ensemble is agnostic to the training algorithm used.

During inference, given a new multivariate time series $X_*$, each model in the ensemble makes a prediction $\hat{y}_j = \sigma(f_j(X_*^j))$ on its respective channel, where $\sigma$ is the softmax (sigmoid) function for converting output logits into a class probability distribution (single probability for binary classification), as is common in ensembles (Abe et al., 2022). The final prediction of the ensemble is obtained by combining the probabilities from all the models in a weighted sum:

$$\hat{y}_{ens} = \sum_{j=1}^{N} w_j \hat{y}_j,$$

where $w_j \in \mathbb{R}^+$ controls the contribution of the prediction from model $f_j$ to the final prediction. In this work, we only consider the uniform ensemble with $w_j = \frac{1}{N}$, though other weighting schemes could be explored in future work.

### 4.2 THEORETICAL ANALYSIS

In this section, we aim to theoretically understand how the channel-wise ensemble impacts OOD generalisation compared to models that learn with CD. Our analysis is based on domain adaptation theory from Ben-David et al. (2006), which provides a bound on the risk in the target domain for a classifier trained on the source domain. We adjust the notation for deep learning models as in Johansson et al. (2019); Chuang et al. (2020).

**Theorem 1** (Ben-David et al. (2006)). *Let $S$ and $T$ be the source and target domains, respectively. With a feature extractor $g$ and hypothesis class $\mathcal{H}$ of classifiers, the hypothesis class for the overall model is $\mathcal{F}(g) = \{h \circ g : h \in \mathcal{H}\}$. With $f$ trained on data from $S$, the risk in $T$ for all $f \in \mathcal{F}(g)$ is:*

$$R_T(f) \leq R_S(f) + d_{\mathcal{H}}(P_S^g(Z), P_T^g(Z)) + \lambda_{\mathcal{F}(g)},$$

*where $R_T = \mathbb{E}_{(X,y) \sim P_T} \ell(f(X), y)$ and $R_S = \mathbb{E}_{(X,y) \sim P_S} \ell(f(X), y)$ are the risk in the target and source domains, respectively, $d_{\mathcal{H}}(P_S^g(Z), P_T^g(Z))$ is the $\mathcal{H}$-divergence between the marginal feature distributions induced by $g$, and $\lambda_{\mathcal{F}(g)}$ is the optimal joint risk that can be achieved from $\mathcal{F}(g)$.*

The bound states that target domain risk is determined by three factors: the source domain risk and two terms that quantify the distribution shift between the source and target domains. From Section

3.1, a shift occurs when $P_S^g(y|Z)P_S^g(Z) \neq P_T^g(y|Z)P_T^g(Z)$, so these terms reflect the difference between the two distributions.

First, the $\mathcal{H}$-divergence term is defined as:

$$d_{\mathcal{H}}(P_S^g(Z), P_T^g(Z)) = \sup_{h \in \mathcal{H}} \left| \Pr_{Z \sim P_S^g}[h(Z) = 1] - \Pr_{Z \sim P_T^g}[h(Z) = 1] \right|,$$

and measures the divergence between the two feature marginal distributions induced by $g$ as how well a classifier from $\mathcal{H}$ can distinguish between features from source and target domain. Second, $\lambda_{\mathcal{F}(g)}$ is defined as:

$$\lambda_{\mathcal{F}(g)} = R_S(f^*) + R_T(f^*), \quad f^* = \inf_{h \in \mathcal{H}} R_S(f) + R_T(f),$$

and measures the difference in the labelling function between the source and target domain, or probabilistically, the conditional label distributions $P_S^g(y|Z)$ and $P_T^g(y|Z)$. As we make the covariate shift assumption, we focus our analyses on the divergence between $P_S^g(Z)$ and $P_T^g(Z)$.

**Extension to the channel-wise ensemble.** We now modify the above bound for the channel-wise ensemble. Using Jensen's inequality, the target domain risk can be expressed as $R_T\left(\sum_{j=1}^N w_j f_j\right) \leq \sum_{j=1}^N w_j R_T(f_j)$, and then as:

$$R_T\left(\sum_{j=1}^N w_j f_j\right) \leq R_S\left(\sum_{j=1}^N w_j f_j\right) + \sum_{j=1}^N w_j d_{\mathcal{H}}(P_S^{g_j}(Z), P_T^{g_j}(Z)) + \sum_{j=1}^N w_j \lambda_{\mathcal{F}(g_j)}. \quad (1)$$

The first term on the right side is the source risk for the ensemble, and the second term is the weighted sum of the *individual-channel* feature marginal distribution divergences.

In our experiments, we report on (approximations of) the source risk and feature marginal distribution divergence to understand how OOD generalisation is impacted by learning with CD and CI. For models that learn with CD, we compute these quantities as defined in Theorem 1. For the channel-wise ensemble, we use Inequality 1.

## 4.3 EXPERIMENTAL SETUP

**Datasets.** We perform experiments on six benchmark multivariate time series classification datasets: DSADS (Altun et al., 2010), HAR (Anguita et al., 2013), MHEALTH (Banos et al., 2014), PAMAP (Reiss & Stricker, 2012), WISDM (Kwapisz et al., 2011) for human activity recognition, and WESAD (Schmidt et al., 2018) for stress and affect detection. Information about each dataset can be found in Appendix A.

**Training and evaluation.** Across all datasets, each participant is defined as a domain. Participants are split equally into four groups, and each group is used as the OOD test set while the remaining three are used for model training. Each reported metric for a dataset-algorithm combination is averaged across the four participant splits. All results are averaged across five runs with different random seeds. For CD, we train a single fully convolutional network (FCN) (Wang et al., 2017) with ERM, and similarly for CI, each model in the ensemble is an FCN trained with ERM. Further training details are provided in Appendix B.

**Metrics.** We report on three metrics based on the analysis in Section 4.2: (1) ID accuracy (as a proxy of source risk) computed on the ID validation set (a subset of $D_{\text{train}}$, see Appendix B.1), (2) $\mathcal{H}$-divergence between source and target domain features, which is approximated with the *proxy-$\mathcal{A}$ distance* (PAD) (Ben-David et al., 2006; Ganin et al., 2016), as described in Appendix C, and (3) OOD accuracy (as a proxy of target risk) computed on $D_{\text{test}}$. We also report the generalisation gap, defined as:

$$\text{Gen. gap} = \text{ID acc.} - \text{OOD acc.},$$

to explicitly measure robustness to distribution shift.

## 4.4 RESULTS

Figure 2 shows each metric for the CD and CI approaches across the six datasets, from which we can make several observations.

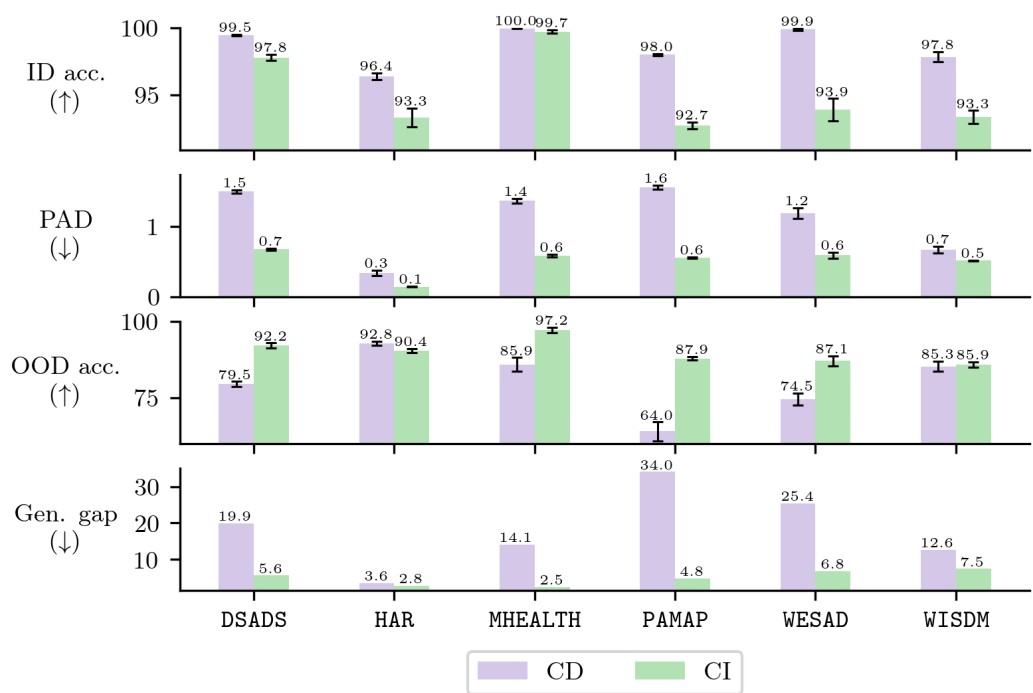

Figure 2: Metrics for CD and CI for each dataset. Error bars represent one standard deviation across five runs. The arrows under each metric shows which direction is better.

**A trade-off between ID accuracy and feature divergence.** Although CI achieves consistently high ID accuracy across all datasets, indicating that learning cross-channel dependencies is not strictly required for effective multivariate TSC, CD consistently outperforms CI. This is consistent with the intuition that richer features of the training data can be learned by considering cross-channel patterns. On the other hand, PAD is lower for CI than CD across all datasets, showing that on average across the ensemble, single-channel models learn features that are more robust to distribution shift than with CD. These findings suggest that a trade-off exists between ID classification performance and feature divergence, which is consistent with Han et al. (2024).

**Implications for OOD generalisation.** Since CD and CI trade off between ID performance and domain divergence, one approach will not *always* be superior for OOD generalisation. Figure 2 shows that CI achieves significantly higher OOD accuracy than CD on the DSADS, MHEALTH, PAMAP, and WESAD datasets. CD exhibits large PAD values on these datasets, and the difference in PAD between CD and CI is substantial. This clearly outweighs the drop in ID performance for CI in determining OOD performance. In contrast, CD outperforms CI in OOD accuracy on HAR. Here, the PAD for CD is far smaller than on the other datasets, and the drop in ID performance for CI becomes more significant. These results suggest that CI is preferred when the distribution shift between domains is more severe, while CD might be preferred when the distribution shift is weaker. In reality, knowing the strength of a distribution shift *a priori* is difficult. The greater reliability offered by CI, evidenced by the smaller generalisation gap across all datasets, might then still be preferred.

**Analysis of individual channels.** Figure 3 shows each metric for each model in the channel-wise ensemble on MHEALTH, with the other five datasets shown in Appendix D.1. The analyses here also apply to the other datasets unless specified. The accuracy metrics reveal that certain channels carry significantly more class-discriminative information than others, and comparing with Figure 2, the performance of the ensemble exceeds the performance of the best single channel member (except for WESAD). Hence, although the models in the ensemble are independent, the late fusion of their predictions makes good use of the information from multiple channels. The PAD for all individual models is lower than the CD model, showing explicitly that single-channel models learn more robust

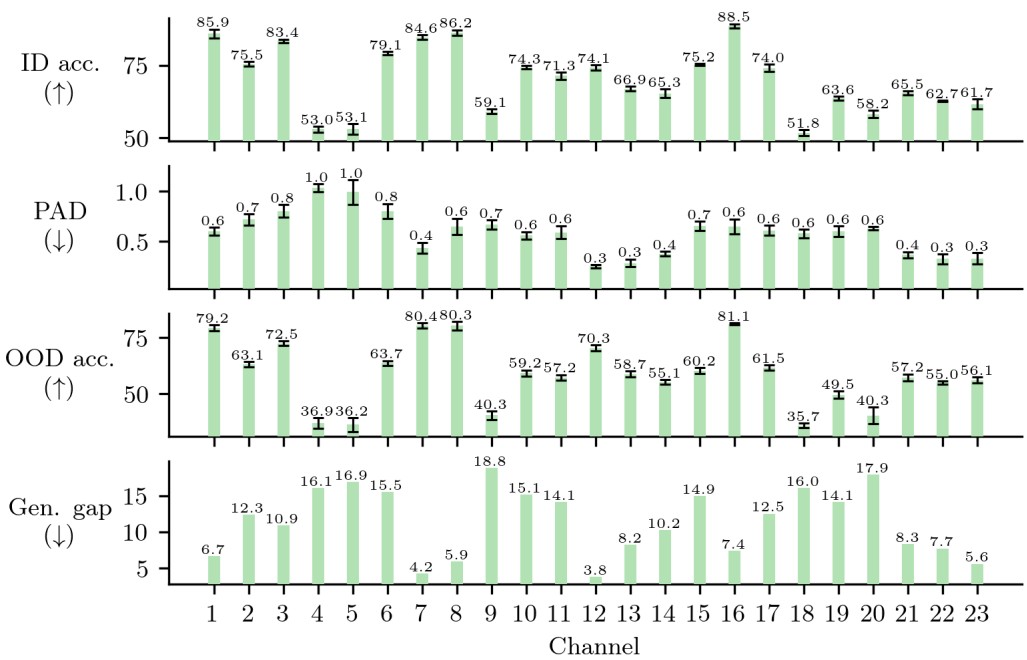

Figure 3: Metrics for each individual member of the channel-wise ensemble on the `MHEALTH` dataset.

features. However, it is worth noting that the generalisation gap is lower for the CD model than some individual models. For models with similar PAD values, such as Channels 16, 17, and 18, a higher ID accuracy typically corresponds to a smaller generalisation gap.

## 5 LEARNING FROM THE FREQUENCY DOMAIN

### 5.1 THE DISCRETE FOURIER TRANSFORM AND FREQUENCY FEATURES

In time series analysis, the discrete Fourier transform (DFT) is used to transform a signal from the time domain, which describes the signal as an amplitude that changes over time, into the frequency domain. For a discrete-time signal $x[n]$ composed of $N$ samples, the DFT, $\mathcal{F}$, and its inverse are defined as:

$$\mathcal{F}\{x[n]\} = X[k] = \sum_{n=0}^{N-1} x[n] e^{\frac{-i2\pi kn}{N}}, \qquad k = 0, \dots, N-1,$$

$$\mathcal{F}^{-1}\{X[k]\} = x[n] = \frac{1}{N} \sum_{k=0}^{N-1} X[k] e^{\frac{i2\pi kn}{N}}, \qquad n = 0, \dots, N-1,$$

where $k$ is the frequency index and $i = \sqrt{-1}$. The DFT decomposes a time domain signal into a sum of complex exponential basis functions at each frequency $k$, each with a coefficient $X[k]$ which may written in exponential form as $X[k] = |X[k]| e^{i\phi}$, where $|X[k]|$ is the amplitude and $\phi$ is the phase:

$$|X[k]| = \sqrt{\mathrm{Re}(X[k])^2 + \mathrm{Im}(X[k])^2}, \qquad \phi = \arctan\left(\frac{\mathrm{Im}(X[k])}{\mathrm{Re}(X[k]))}\right),$$

where $\mathrm{Re}(\cdot)$ and $\mathrm{Im}(\cdot)$ are the real and imaginary parts of a complex number, respectively.

**Feature learning.** Mohapatra et al. (2024) showed that using separate feature extractors for the magnitude and phase of the DFT coefficients, followed by late feature fusion, outperforms joint encoding (concatenating them first) or using a single modality. Following this approach, we extract

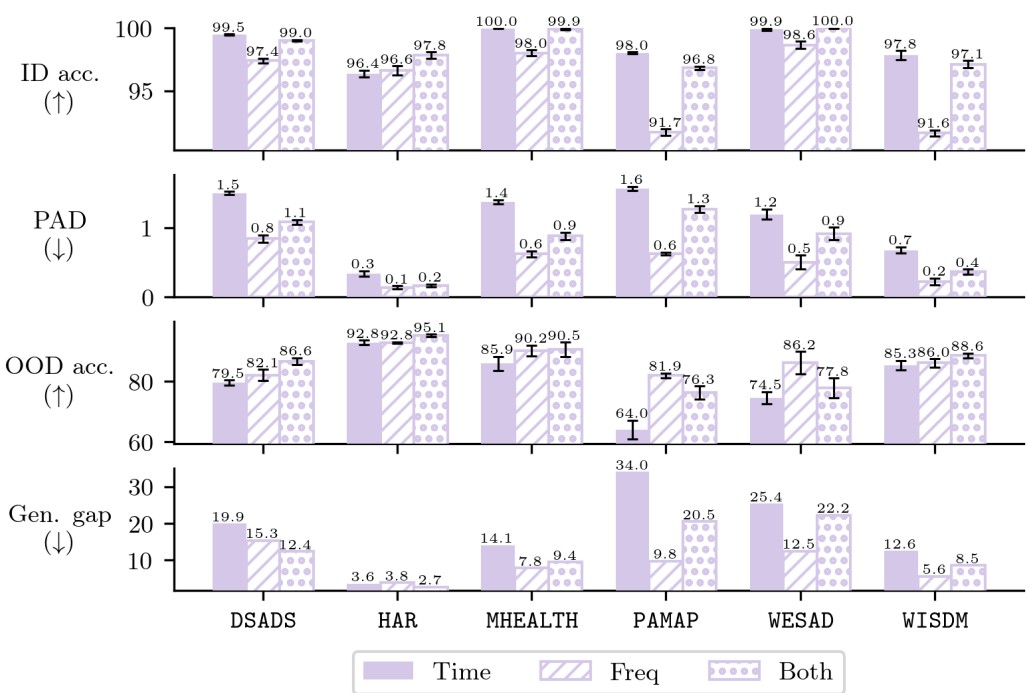

Figure 4: Metrics for CD with time, frequency, and both features.

separate magnitude and phase feature vectors. We then concatenate them and project to a lower dimensional space (equal to that of the time feature vector) using a linear layer, forming the overall frequency feature vector. We concatenate time and frequency feature vectors before classification as in He et al. (2023).

## 5.2 RESULTS

In these experiments, the setup is the same as before (Section 4.3). We show results for models with either time features or frequency features to understand their behaviour individually, or both as described above.

**Frequency features improve OOD generalisation with CD.** Figure 4 shows that frequency features consistently yield lower PAD but also lower ID accuracy (except for `HAR`) compared to time features. When time and frequency features are concatenated, ID performance is preserved while distributional divergence is reduced, leading to improved OOD accuracy across all datasets compared to time features alone. This supports He et al. (2023) in showing that frequency domain features improve OOD generalisation in TSC, *when the model learns with CD*. Comparing with Figure 2, it is important to note that in terms of OOD accuracy and generalisation gap, this strategy is still inferior to using CI with time features on datasets where the distribution shift is significant.

**Methods to improve OOD generalisation are architecture-specific.** Figure 5 shows the same trend in ID accuracy and PAD for CI as with CD. However, the effect that this has on OOD performance is different, with no significant benefit being observed by using frequency features (except on `WISDM`, where ID performance is improved by using both). Our explanation for this is that the domain divergence for time features is already low, and the drop in ID performance outweighs any gain from further reducing divergence. We do not mean to imply here that frequency features are not useful for improving OOD generalisation when using CI, as other architectures may use those features in a more effective way. But rather we highlight that a strategy that helps to improve OOD generalisation for models that learn with CD might not help models that learn with CI.

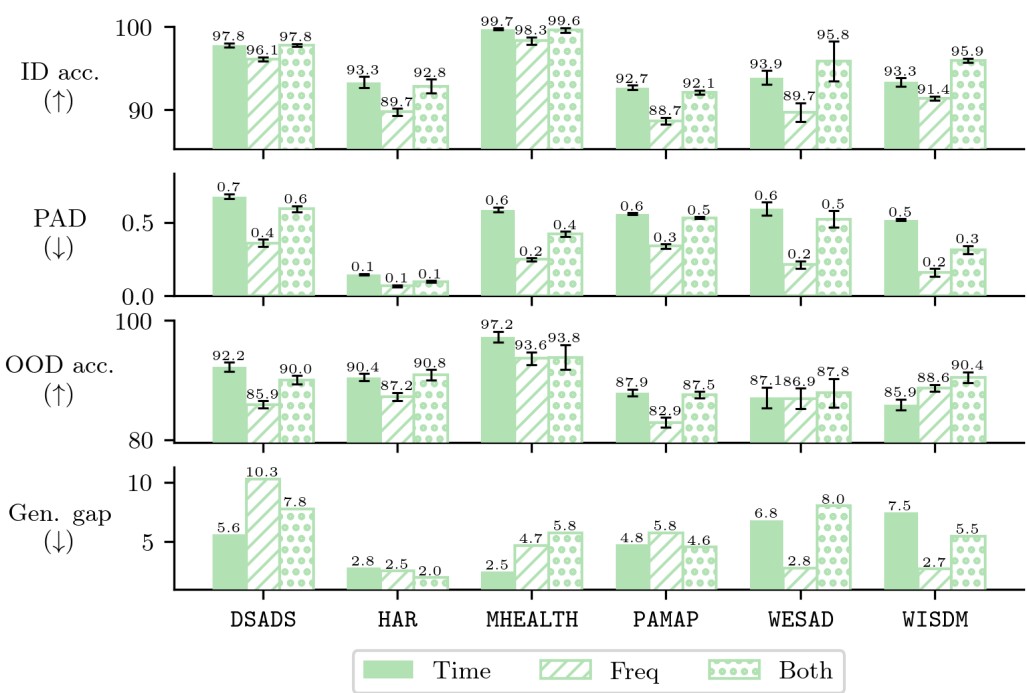

Figure 5: Metrics for CI with time, frequency, and both features.

## 6 CONCLUSIONS

In this paper, we have investigated how learning with CI impacts OOD generalisation in multivariate TSC, and compared it with the standard approach of learning with CD. We have shown that CI significantly improves robustness to distribution shift, as evidenced by smaller feature divergences and generalisation gaps across all the six benchmarks we used. For datasets where the distribution shift is more severe, this improved robustness vastly improves classification performance on OOD data. Below, we present limitations of our work and highlight some potential future research avenues.

**Learning channel dependencies.** The CI approach assumes that learning dependencies between channels is not strictly necessary for good performance. Although our experiments suggest that this assumption is reasonable, it is entirely possible that a dataset might require learning these dependencies for effective classification, in which case the approach would fail. Therefore, a natural extension to this research would be to explore how both types of learning might be incorporated into a single solution.

**Improving performance for CI.** Our results on using frequency features show that to improve OOD generalisation further for the channel-wise ensemble, an approach that trades off ID classification performance for robustness will likely not be effective. Instead, approaches will need to either (1) maintain ID classification performance while improving robustness or (2) maintain robustness while improving ID classification performance.

**Beyond time series classification.** Finally, we recognise that our methods and findings are specific to multivariate TSC, but we hope that they might inspire those investigating OOD generalisation in other domains to take a step back and consider all the assumptions and choices, perhaps taken for granted, that could be impacting OOD generalisation. This might help to uncover other simple techniques for improving OOD generalisation.

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

## A  DATASETS

In this section, we provide details for each of the benchmark datasets that we used in this paper. To obtain multiple input samples from a single continuous recording, a sliding window approach is used, whereby a window of fixed size is passed along the time series, creating segments at fixed intervals. If the class labels for each time step of the window are all the same, the window is used, and is assigned that class label. This prevents windows being used that are assigned a class label but contain individual time steps that have been recorded as a different label.

**DSADS** (Altun et al., 2010). This dataset contains data from eight subjects for the task of human activity recognition, with 19 classes. The data was recorded with a 3-axis accelerometer, 3-axis gyroscope, and 3-axis magnetometer device at five different locations on the body (total 45 channels), with a sampling frequency of 25 Hz. The dataset is already segmented into windows of 125 samples (five seconds). The average number of samples for each domain is 1140.

**HAR** (Anguita et al., 2013). This dataset contains data from 30 subjects for the task of human activity recognition, with six classes: walking, walking upstairs, walking downstairs, sitting, standing, and laying. The data was recorded with a 3-axis accelerometer, 3-axis gyroscope, and 3-axis body device (total nine channels), with a sampling frequency of 50 Hz. The dataset is already segmented into windows of 128 samples (2.56 seconds). The average number of samples for each domain is 343.

**MHEALTH** (Banos et al., 2014). This dataset contains data from 10 subjects for the task of human activity recognition, with 12 classes. The data was recorded with a 3-axis accelerometer, 3-axis gyroscope, and 3-axis magnetometer device at two different locations on the body, and a third device on the chest with a 3-axis accelerometer and two-lead ECG (total 23 channels), with a sampling frequency of 50 Hz. We segment the dataset with a window length of 100 samples (two seconds) and overlap of 50 samples. The average number of samples for each domain is 663.

**PAMAP** (Reiss & Stricker, 2012). This dataset contains data from nine subjects for the task of human activity recognition, with 18 classes. The data was recorded with a temperature sensor, 3-axis accelerometer, 3-axis gyroscope, 3-axis magnetometer device at three different locations on the body (total 30 channels), with a sampling frequency of 100 Hz. We segment the dataset with a window length of 256 samples (2.56 seconds) and overlap of 128 samples. The average number of samples for each domain is 820.

**WESAD** (Schmidt et al., 2018). This dataset contains data from 15 subjects for the task of stress and affect detection with two classes: stressed and non-stressed. The data was recorded with an ECG, EDA, EMG, respiration, and temperature sensor and 3-axis accelerometer (total 8 channels), with a sampling frequency of 700 Hz. We downsample the signals to 100 Hz to ease computational costs. We segment the dataset with a window length of 6000 samples (60 seconds) and overlap of 1000 samples. The average number of samples for each domain is 172.

**WISDM** (Kwapisz et al., 2011). This dataset contains data from 36 subjects for the task of human activity recognition, with six classes: walking, jogging, walking upstairs, walking downstairs, sitting, and standing. The data was recorded with a 3-axis accelerometer, with a sampling frequency of 20 Hz. We segment the dataset with a window length of 128 samples (6.4 seconds) and no overlap. The average number of samples for each domain is 228.

# B    ADDITIONAL EXPERIMENTAL DETAILS

## B.1    DATA SPLIT

We use the 'training-domain validation set' model selection strategy (Gulrajani & Lopez-Paz, 2021). Here, the dataset $D_d$ from a domain $d \in \mathcal{E}_{\text{train}}$ with indices $\mathcal{I}_d = \{1, \ldots, n_d\}$ is split into a training set with indices $\mathcal{I}_d^{\text{train}} \subset \mathcal{I}_d$ and a validation set with indices $\mathcal{I}_d^{\text{val}} = \mathcal{I}_d \backslash \mathcal{I}_d^{\text{train}}$, such that we have a training set $D_d^{\text{train}} = \{(X_i, y_i) | (X_i, y_i) \in D_d \text{ and } i \in \mathcal{I}_d^{\text{train}}\}$ and a validation set $D_d^{\text{val}} = \{(X_i, y_i) | (X_i, y_i) \in D_d \text{ and } i \in \mathcal{I}_d^{\text{val}}\}$. The training and validation sets from each domain are combined to form the overall training set $D_{\text{train}} = \cup_{d \in \mathcal{E}_{\text{train}}} D_d^{\text{train}}$ and validation set $D_{\text{val}} = \cup_{d \in \mathcal{E}_{\text{train}}} D_d^{\text{val}}$. We use 75% of the data in $D_d$ for training and 25% for validation, stratified by class.

## B.2    MODEL TRAINING

We use a fully convolutional network (Wang et al., 2017) as the feature extractor for all experiments. It consists of three 1D convolutional layers, each with 16 kernels of length 3, with batch normalisation and the ReLU activation function between layers. Global average pooling is used on each feature map from the last layer to obtain the final 16-dimensional feature vector. For classification, the feature vector is passed to a linear classifier.

All models are trained for 50 epochs using the Adam optimiser with a learning rate of $1 \cdot 10^{-3}$, weight decay of $1 \cdot 10^{-5}$, and a batch size of 64. For recording metrics, we select the epoch with the lowest validation loss.

All experiments were implemented in PyTorch and ran on a single NVIDIA GeForce RTX 3090.

# C    PROXY-$\mathcal{A}$ DISTANCE

We approximate the $\mathcal{H}$-divergence with the proxy-$\mathcal{A}$ distance (PAD), following the approach of Ganin et al. (2016). This first involves creating a new dataset of source domain feature vectors from $D_{\text{train}}$, each labelled as 0, and a dataset of target domain feature vectors from $D_{\text{test}}$, each labelled as 1. The larger dataset is truncated such that they are both the same size $N$, and half of each dataset is then randomly selected as the train split, and the other half as the test split:

$$U_{\text{train}}^S = \{(Z_i, 0)\}_{i=1}^{N/2}, \ U_{\text{test}}^S = \{(Z_i, 0)\}_{i=(N/2)+1}^N,$$
$$U_{\text{train}}^T = \{(Z_i, 1)\}_{i=1}^{N/2}, \ U_{\text{test}}^T = \{(Z_i, 1)\}_{i=(N/2)+1}^N.$$

They are combined into the overall training and test datasets:

$$U_{\text{train}} = \{U_{\text{train}}^S \cup U_{\text{train}}^T\}, \quad U_{\text{test}} = \{U_{\text{test}}^S \cup U_{\text{test}}^T\}.$$

A two-layer MLP is trained on $U_{\text{train}}$ to distinguish between source and target domain features, and then tested on $U_{\text{test}}$. The mean absolute error is computed for the predictions to obtain the error $\epsilon$. The PAD is then computed as:

$$\text{PAD} = 2(1 - 2\epsilon).$$

# D    ADDITIONAL RESULTS

## D.1    INDIVIDUAL CHANNEL RESULTS

In these figures, we show the four metrics for the individual ensemble members for the other five datasets (i.e. not MHEALTH).

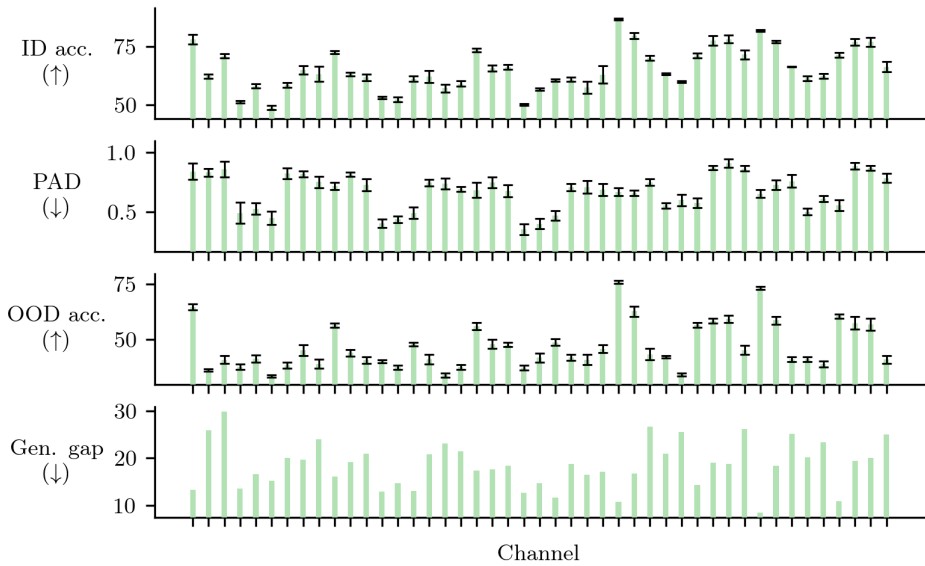

Figure 6: `DSADS`. Individual values are not shown because of legibility with the large number of channels.

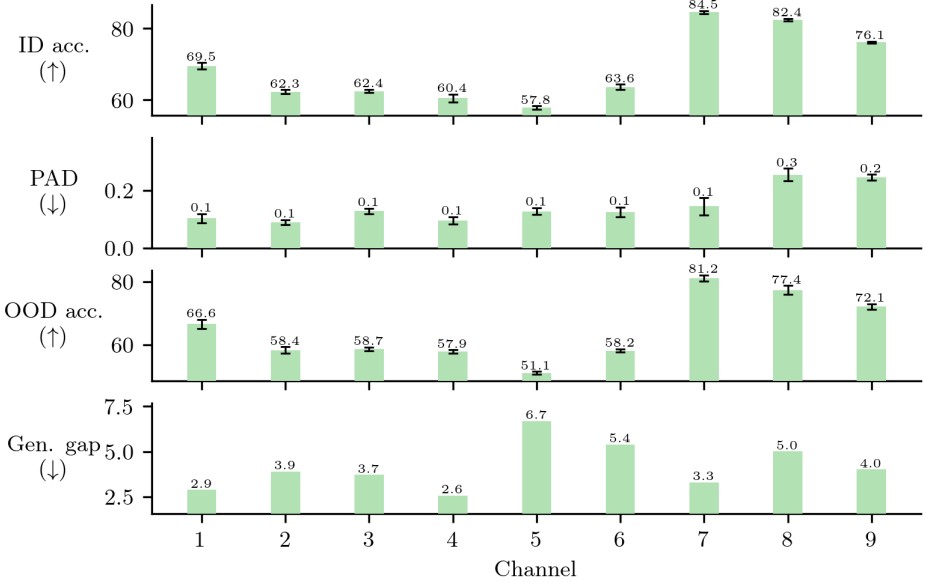

Figure 7: `HAR`.

# E    MISCELLANEOUS

## E.1    FREQUENCY DOMAIN WINDOWING

The discrete Fourier transform (DFT) treats a signal being analysed as if it is an integer number of periods of a periodic signal. However, as described in Appendix A, individual time series to be classified are obtained by segmenting a longer time series, which can create discontinuities at the boundaries of the segmented signal, i.e. an abrupt change from $x[N-1]$ to $x[0]$. This results in *spectral leakage* and a worsening of the quality of the frequency spectrum. To remedy this, a window function can be used to enforce continuity at the boundaries of the signal. We apply a

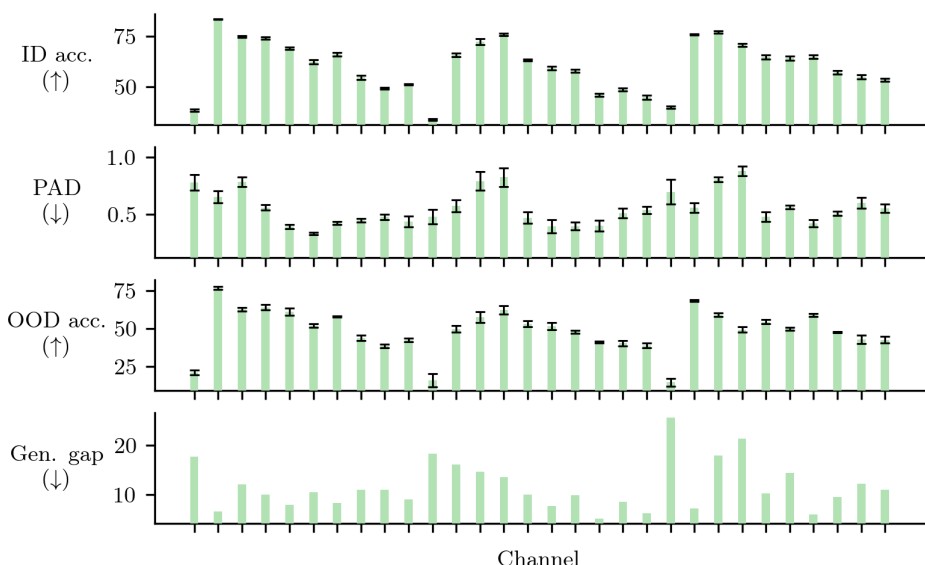

Figure 8: PAMAP. Individual values are not shown because of legibility with the large number of channels.

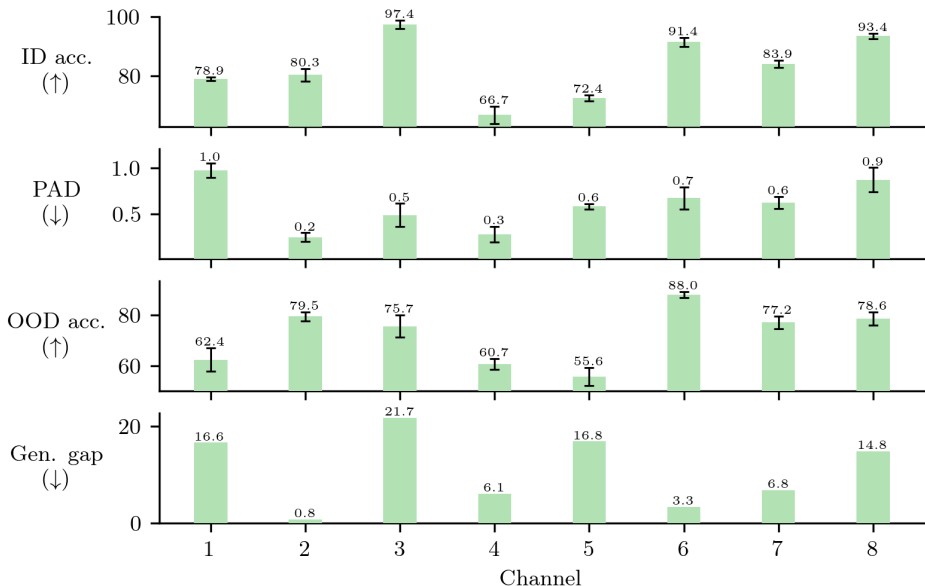

Figure 9: WESAD.

cosine window, defined as $w[n] = \sin\left(\pi(n + 0.5)/N\right)$, to each channel of each time series before computing the DFT.

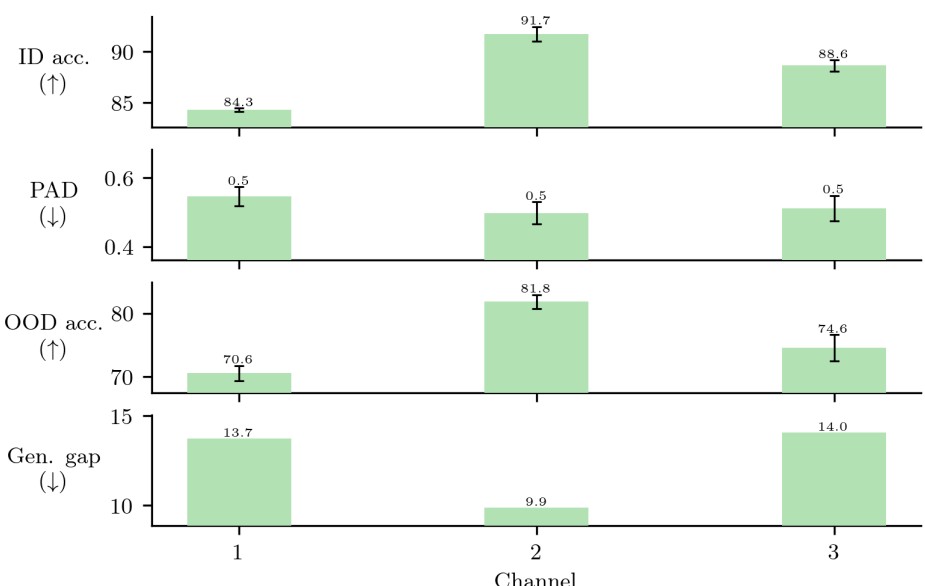

Figure 10: `WISDM`.

