# OpenReview forum: "Channel Independence Improves Out-of-Distribution Generalisation in Multivariate Time Series Classification"
_ICLR.cc/2025/Conference — ICLR 2025 Conference Withdrawn Submission_

### Official Review · Reviewer_rT8V · 2024-10-31

**Soundness:** 3
**Presentation:** 2
**Contribution:** 2
**Rating:** 5
**Confidence:** 3

**Summary:**

This paper argues that the ongoing discussion of CI v. CD learning in the context of time series regression may also have interesting results in the case of time series classification.  The work then demonstrates that a large generalization gap occurs across six real-world human-activity datasets under a particular type of OOD shift.  It is demonstrated how this robustness gap can often lead to better performance with the (less powerful) CI models.  Finally, some theory is discussed as an explanation for why this robustness gap is occurring.

**Strengths:**

The work applies to a relatively lesser explored application of time series classification which extends the existing discussion of channel independence and channel dependence which is currently focused on the case of regression.

The work achieves the expected result of better robustness for simpler models.

Figures displaying the results make it clear to digest the real-world experiments done across six real-world datasets.

Figure 3 begins to give insights into how the individual features may be heterogeneously used by the model, which may be easier to do in the simpler CI space when compared to the CD space.

**Weaknesses:**

When transitioning form TSR to TSC, the specific nuances of the classification regime are not also considered and the existing CI v. CD distinction from TSR is exactly copied into the TSC regime.  In particular, in line 120-122, the authors introduce a mixed CI/CD approach which extracts CI features which are then combined with another MLP (CD) for final classification.  This is a regime which is not possible in the TSR regime and is seemingly left completely unexplored in this work.

The time domain features and frequency domain features are both included but very little analysis is given.  Importantly, the major conclusion that CI is generally better than CD does not hold for the results in the frequency space.  This is not sufficiently discussed in the current version of the work.

Only one model is used as a representative for all CI models and all CD models.  It is unclear how straightforward this distinction is and how robust the results are to different architectures.  Given the above change in feature landscape has a significant impact, it should be expected there may also be sensitivity to slight architecture changes.

It is unclear how generically the results are for all of TSC.  Some key factors in the empirical evaluation could be potentially identified as limitations.  In particular, all six datasets correspond to human trajectories, which may not be representative of all TSC tasks, and second the distribution shift is always corresponding to the shift in human which was taken as the generator of the trajectory.  This is a specific type of OOD shift which may be stronger as well as more specific than a general OOD shift.  The theoretical results do not seem strong enough to support the genericness of such claims.

**Questions:**

Can you clarify if in your analysis you were able to uncover any key features of the datasets where "CI-time" outperformed CD-time" as well as when "CI-freq" outperformed "CD-freq"? How do you think modified architectures like [1] would fit into this dichotomy?

The most important quantity practically speaking is not the generalization gap, but rather the final performance.  From this lens, it could be argued that this work has made minimal progress towards understanding the question of whether to use CI or CD in TSC.  How do you feel your theoretical results and empirical results support an answer to the question of whether to use CI or CD in the domain of TSC?

In this work, all results seem to be stated for a CNN architecture.  Do you also use CNNs for the frequency domain?  How well do you think these results will generalize to other architectures and why?


[1] "Time Series Classification Using Multi-Channels Deep Convolutional Neural Networks" Yi Zheng, et al. 2014.

---

### Official Review · Reviewer_scfK · 2024-11-01

**Soundness:** 2
**Presentation:** 3
**Contribution:** 1
**Rating:** 3
**Confidence:** 5

**Summary:**

This manuscript proposes to use the channel independence (CI) method in the time series data classification problem, and conducts theoretical analysis, showing that the CI method can improve the generalization ability of the model for out-of-distribution data. The authors also conduct relevant experimental analysis, showing that the effect of CI is better than CD. Moreover, the authors conduct relevant experiments on frequency domain analysis, and the results also show that CI can enhance the generalization ability of the model.

**Strengths:**

1. The motivation of this study is very clear. It applies the CI method, which performs well in time series prediction tasks, to time series classification tasks, and achieves significant improvement in results.
2. The authors design and conduct a variety of experiments, with clear experimental methods and credible results.

**Weaknesses:**

1. The research lacks innovation and novelty, and it only transplants the CI method (arXiv preprint arXiv:2211.14730, 2022.) in the prediction task to the classification task. This change is too simple and lacks sufficient academic value.
2. The author's theoretical analysis seems to be mainly based on the related work of (Shai Ben-David, John Blitzer, Koby Crammer, and Fernando Pereira. Analysis of representations for domain adaptation. In NeurIPS, 2006.), lacking independent research contributions.

**Questions:**

1. How can the datasets used by the study reflect the generalization ability of the model to “out of distribution” data? Can the authors explicitly show the out-of-distribution data characteristics of the relevant datasets?

---

### Official Review · Reviewer_ZEwE · 2024-11-04

**Soundness:** 2
**Presentation:** 3
**Contribution:** 2
**Rating:** 5
**Confidence:** 4

**Summary:**

This paper addresses the critical issue of model robustness to distribution shifts in machine learning, particularly within the context of multivariate time series classification (TSC). The authors investigate the impact of channel independence (CI) on out-of-distribution (OOD) generalization and compare it with the conventional approach of channel dependence (CD). Through experiments on six real-world multivariate time series datasets, this paper demonstrates that models employing CI exhibit smaller distributional divergence and thus are more robust to distribution shifts, leading to improved OOD accuracy, especially on datasets with more severe distribution shifts.

**Strengths:**

* This paper tackles a significant challenge in machine learning—OOD generalization—which is crucial for the safe and effective deployment of models in real-world applications. The exploration of CI in TSC is a novel approach that offers fresh insights into improving model robustness.

* This paper is supported by empirical evidence from six diverse real-world datasets.

* This paper further explores the impact of frequency domain features on OOD generalization within the context of CD and CI, by comparing the distributional differences and classification performance between time domain and frequency domain features, which provides a more comprehensive perspective.

* This paper is well-organized and clearly written, making complex concepts accessible and the findings easy to follow.

**Weaknesses:**

* This paper's contribution is somewhat marginal; it focuses more on defining and elucidating the problem without theoretical bounds on the generalization, with less advancement in theory and algorithms, hence its impact on the field of multivariate time series classification is not particularly significant.

* The experiments in the paper are relatively weak, which does not strongly support the conclusions, and the use of only one model and a limited number of datasets weakens the robustness of the findings. In detail, experiments on more datasets covering different types of relationships among variables (such as spatial dependencies, mutual influences, etc) could make the conclusion in this paper more convincing.

**Questions:**

*  The experimental results in the paper are based solely on a 1D-CNN classification model, which seems somewhat limited. What will happen if we apply non-CNN multivariate time series classification methods, such as those based on transformers (e.g., shapeformer, SVP-T) and contrastive learning methods (Ts-vec)?

* There are six datasets used in this paper, which is insufficient to prove the universality of channel independence. In particular, the UCR (UEA) dataset is a standard choice for many multivariate time series classification methods, and it would be interesting to see how the paper's approach fares on this benchmark. In detail, the benchmark datasets used in the article, with the exception of WESAD, all pertain to human activity recognition. Therefore, I suggest incorporating datasets from other domains within the UEA, such as those for motion classification, ECG classification, EEG/MEG classification, and audio spectra classification. Additionally, as mentioned in the survey (Deep learning for time series classification and extrinsic regression: a current survey. ACM Computer Surveys, 2024.), datasets related to earth observation satellites and instruments could be utilized.

* Could the authors elaborate on the criteria used for selecting the window lengths and the decision-making process behind the use of overlap in the context of this study?

* Given the proximity of anomaly detection tasks to classification tasks, it would be valuable to have experimental results that demonstrate how the paper's conclusions apply to anomaly detection scenarios. Are there any additional experiments or analyses that could provide insights into this aspect? Four benchmark datasets widely used in multivariate time series anomaly detection research include SWaT, SMD, SMAP, and MSL. The extension of different methods and different downstream tasks can make the value of this article more significant.

---

### Official Review · Reviewer_7Tc3 · 2024-11-04

**Soundness:** 1
**Presentation:** 2
**Contribution:** 1
**Rating:** 3
**Confidence:** 4

**Summary:**

This paper investigates the impact of Channel Independence (CI) and Channel Dependence (CD) on Out-of-Distribution (OOD) generalization in time series classification (TSC). The authors propose a channel-wise ensemble method that leverages the advantages of CI to effectively handle multivariate TSC problems. They compare this method with CD-based approaches through theoretical analysis and experiments on six real-world datasets. In addition, they analyze the performance of combining time-domain and frequency-domain features to address distribution shifts. The results indicate that while CD performs better on in-distribution (ID) data, CI offers superior OOD generalization capabilities, demonstrating high robustness, especially under significant distribution shifts. Introducing frequency features also improves OOD performance in CD models but does not provide additional benefits in CI models.

**Strengths:**

1. The paper provides a well-articulated motivation for exploring the impact of CI and CD on OOD generalization in TSC, addressing a significant challenge in deploying time series machine learning models in real-world applications.
2. By employing domain adaptation theory, the authors offer a theoretical explanation for why CI may generalize more robustly under certain conditions, grounding their experimental findings.

**Weaknesses:**

1. The theoretical analysis using domain adaptation theory seems to lack depth and does not connect well to the proposed method. For example, in Section 4.2, the upper bound from Theorem 1 is approximated using Jensen's inequality without discussing the tightness of this bound. In fact, the theoretical inequality applied to CD in the experiment and the theoretical inequality of CI are inequalities that, as a result, have different upper bounds. This omission raises questions about the validity of the theoretical conclusions drawn. For another example, the assumption that distribution shift is solely due to covariate shift ($P_S(X) \neq P_T(X)$ and $P_S(y|X) = P_T(y|X)$) may not hold in real-world scenarios. The paper does not provide empirical evidence to support this assumption within their experimental setup.

2. The methodological positioning and contribution are also confused or weak. For example, the authors label their method as a CI method, but it differs from most existing CI methods in the time series forecasting literature. CI methods, like DLinear, use a "shared backbone" applied independently to different channels. In contrast, the proposed method involves training separate models for each channel and combining them in an ensemble. This could cause confusion and may misrepresent the novelty of the approach. References: [1], [2], [3]. For another example, the channel-wise ensemble approach resembles existing methods in time series classification, such as those discussed by Ruiz et al. (2021) and the HIVE-COTE ensemble methods. The paper does not clearly differentiate its contributions from these existing works, nor does it provide direct comparisons. References: [4], [5], [6].

[1] Han, L., Ye, H. J., & Zhan, D. C. (2024). The capacity and robustness trade-off: Revisiting the channel independent strategy for multivariate time series forecasting. IEEE Transactions on Knowledge and Data Engineering.

[2] Zeng, A., Chen, M., Zhang, L., & Xu, Q. (2023, June). Are transformers effective for time series forecasting?. In Proceedings of the AAAI conference on Artificial Intelligence (Vol. 37, No. 9, pp. 11121-11128).

[3] Nie, Y., Nguyen, N. H., Sinthong, P., & Kalagnanam, J. (2022). A time series is worth 64 words: Long-term forecasting with transformers. arXiv preprint arXiv:2211.14730.

[4] Ruiz, A. P., Flynn, M., Large, J., Middlehurst, M., & Bagnall, A. (2021). The great multivariate time series classification bake off: a review and experimental evaluation of recent algorithmic advances. Data Mining and Knowledge Discovery, 35(2), 401-449.

[5] Bagnall, A., Flynn, M., Large, J., Lines, J., & Middlehurst, M. (2020). On the usage and performance of the hierarchical vote collective of transformation-based ensembles version 1.0 (hive-cote v1. 0). In Advanced Analytics and Learning on Temporal Data: 5th ECML PKDD Workshop, AALTD 2020, Ghent, Belgium, September 18, 2020, Revised Selected Papers 6 (pp. 3-18). Springer International Publishing.

[6] Middlehurst, M., Large, J., Flynn, M., Lines, J., Bostrom, A., & Bagnall, A. (2021). HIVE-COTE 2.0: a new meta ensemble for time series classification. Machine Learning, 110(11), 3211-3243.

3. The validation part is also confused or weak. For example, the authors define each participant as a domain but do not verify whether genuine distribution shifts exist between the training and test groups. Statistical analysis demonstrating the presence and extent of distribution differences is necessary to substantiate the claims about OOD generalization. For another example, all six datasets are related to human activity recognition or stress detection. This narrow focus may limit the generalizability of the findings to other domains, such as finance or environmental monitoring.

4. Other points may be considered weak. For example, the paper does not compare the proposed method with other state-of-the-art OOD generalization techniques. Without such comparisons, it is challenging to evaluate the method's relative performance and contributions. For another example, the ensemble assigns equal weights to all channel models. The potential benefits of alternative weighting schemes, such as weighting based on individual channel performance, are not explored. Meanwhile, the experiments are conducted exclusively with Fully Convolutional Networks (FCNs). It remains unclear whether the observed benefits of CI extend to other architectures like RNNs or Transformers.

5. Finally, while the authors argue that there is not much work on the OOD generalization of the TSC task, they did not pinpoint the exact reason why CI is advantageous over CD in the context of TSC. They just borrow theories and concepts that are non-specific to the TSC task. based on the above concerns, particularly regarding the misrepresentation of the method, insufficient theoretical analysis, lack of verification of distribution shifts, and inadequate comparison with existing methods, I recommend that this paper be rejected in its current form. Addressing these issues could significantly strengthen the work for future submission.

**Questions:**

Please see the weaknesses.

1. For example, how does your method fundamentally differ from traditional CI approaches that use shared backbones independently on each channel? What is the rationale for labeling your method as CI, and why can't existing CI methods be directly applied in your context?

2. For another example, can you please provide more insight into the tightness of the upper bound approximated using Jensen's inequality? How does this approximation impact the validity of your theoretical analysis and the comparison between CD and CI?

3. Meanwhile, have you conducted statistical analyses to confirm that the domains (participants) exhibit significant distribution shifts? Providing evidence of genuine distribution differences would strengthen your claims about OOD generalization.

4. Finally, can you please validate that your findings can generalize to datasets from other domains beyond human activity recognition, such as financial or transportation time series data?

---

### Note · Authors · 2024-11-21

**Comment:**

After careful consideration, we have decided to withdraw our paper. We greatly appreciate the time and effort the reviewers dedicated to evaluating our work. The feedback provided is highly constructive and will help us refine and strengthen our research moving forward.

**Withdrawal Confirmation:**

I have read and agree with the venue's withdrawal policy on behalf of myself and my co-authors.